# Study on the Deposition Characteristics of Molybdenum Thin Films Deposited by the Thermal Atomic Layer Deposition Method Using MoO$_2$Cl$_2$ as a Precursor

**Baek-Ju Lee \*, Kyu-Beom Lee, Min-Ho Cheon, Dong-Won Seo and Jae-Wook Choi**

Vacuum Equipment R&D Division, Semiconductor Research Center, Hanwha Corporation, Pangyo-ro 305, Seongnam 13488, Republic of Korea; kyu-beom.lee@hanwha.com (K.-B.L.); minhocheon@hanwha.com (M.-H.C.); dw.seo@hanwha.com (D.-W.S.); jw_choi@hanwha.com (J.-W.C.)
\* Correspondence: baekju.lee@hanwha.com

**Abstract:** In this study, we conducted research on manufacturing molybdenum (Mo) thin films by a thermal atomic layer deposition method using solid MoO$_2$Cl$_2$ as a precursor. Mo thin films are widely used as gate electrodes and electrodes in metal-oxide semiconductor field-effect transistors. Tungsten (W) has primarily been used as a conventional gate electrode, but it suffers from reduced resistivity due to the residual fluorine component generated from the deposition process. Thus, herein, we developed a Mo thin film with low resistivity that can substitute W. The MoO$_2$Cl$_2$ precursor used to deposit the Mo thin film exists in a solid state. For solid precursors, the vapor pressure does not remain constant compared to that of liquid precursors, thereby making it difficult to set process conditions. Furthermore, the use of solid precursors at temperatures 600 °C and above has many limitations. Herein, H$_2$ was used as the reactive gas for the deposition of Mo thin films, and the deposition temperature was increased to 650 °C, which was the maximum processing temperature of the aluminum nitride heater. Additionally, deposition rate, resistivity change, and surface morphology characteristics were compared. While resistivity decreased to 12.9 μΩ·cm with the increase of deposition temperature from 600 °C to 650 °C, surface roughness (Rq) was increased to 0.560 nm with step coverage of 97%. X-ray diffraction analysis confirmed the crystallization change in the Mo thin film with increasing process temperature, and a certain thickness of the seed layer was required for nucleation on the initial wafer of the Mo thin film. Thus, the molybdenum nitride thin film was deposited after the 4 nm deposition of Mo thin film. This study confirmed that crystallinity of Mo thin films must be increased to reduce their resistivity and that a seed layer for initial nucleation is required.

**Keywords:** MoO$_2$Cl$_2$; resistivity; metal ALD; thermal ALD; XRD

## 1. Introduction

Deposition is a method of coating a thin solid film of several micrometers onto the surface of an object made from metal or plastic with metal particles in a gaseous state. In terms of covering the surface of the object with a metal film, it can be considered to be similar to plating, but the difference is in the state of the metal moving to the surface of the object. Unlike the general plating process in which an object is immersed in a solution such as an electrolyte in which metal is dissolved, and deposited with a metal film, deposition creates a thin film with vaporized metal. Therefore, unlike the plating process, there is an advantage in that there is no use of harmful chemicals or treatment of waste organic matter. Deposition is divided into Chemical Vapor Deposition (CVD) and Physical Vapor Deposition (PVD) according to the principles of creating metal vapor. CVD is divided according to process pressure, state of injection source, energy source, etc. PVD has a classification method according to the method of forming metal vapor (Table 1).

**Table 1.** Type of deposition.

| Fundamental | Method | Energy Source |
|---|---|---|
| Physical | PVD | Sputtering |
| | | Evaporation |
| | Spin-on | Liquid precursor |
| Chemical | CVD | APCVD |
| | | LPCVD |
| | | PECVD |
| | ALD | PEALD |
| | | Thermal ALD |
| | Plating | Electroplating |
| | | Electroless plating |

PVD is a method in which the energy applied to a desired metal material is converted into kinetic energy, and the material moves and accumulates on a substrate to form a thin film. PVD has a thickness of several nanometers to several thousand nanometers and can be deposited regardless of the size of the substrate. PVD is divided into thermal evaporation vacuum deposition, the sputtering method and the ion assisted method according to the gas generation method. CVD is a method in which elements in a source gas in the process chamber are chemically transformed into other elements that adhere to and deposit on the surface of a wafer. CVD is one of the earliest semiconductor processes and has evolved substantially over time. Methods that use thermal energy include atmospheric pressure CVD and low pressure CVD, and methods that use plasma energy include plasma enhanced CVD (using low density plasma) and high density plasma CVD. Recently, however, the usage of atomic layer deposition (ALD) has been increasing and active studies have been conducted domestically and internationally for manufacturing next-generation semiconductors that integrate an increasing number of complex circuits in smaller units. A self-limiting reaction is caused by the chemical adsorption on the surface, which results in deposition of Å-level thin films individually, allowing precise thickness control. In addition, it is an essential deposition technology for next-generation semiconductor manufacturing because its step coverage characteristics in fine patterns are superior to CVD [1,2]. The ALD deposition technology is being explored vigorously in research for its use in the development of miniaturized semiconductors and perovskite solar cells. Originally employed to deposit insulating layers to control the flow of electrons, the application of this technology has recently expanded to metal wiring processes as well. As a next-generation semiconductor material, molybdenum (Mo) thin films possess excellent physical properties of extremely high melting point, low thermal expansion coefficient, low resistivity, and high thermal conductivity. Consequently, it is increasingly used to manufacture semiconductor devices for diffusion barriers, electrodes, photomasks, power electronic device substrates, and low-resistivity gates and connectors. These physical properties have motivated researchers to achieve the deposition of Mo thin films characterized by high conformality and fast deposition rates tailored to efficient mass fabrication processes. Because of their low resistivity, transition metals, such as Mo, are widely used as metal electrodes in solar cells or semiconductors, where energy loss must be minimal [3–5]. Particularly, Mo thin films are highly utilized as electrodes in the gate and metal-oxide-semiconductor field-effect transistors (MOSFETs). A $WF_6$ precursor is used to deposit tungsten, which is a traditional metal gate material. However, using $WF_6$ can leave a fluorine substance on the wafer, which can affect semiconductor performance. Tungsten used in metal wiring usually has a low resistance. However, if there is any foreign impurity, there is a risk of an increase in its resistance value, and fluorine can cause etching by its nature [6,7]. To solve this problem, Mo has been actively studied as a substitute for tungsten. Existing

research on Mo thin films has focused on chemical vapor deposition (CVD) methods via Mo-containing precursors and a reaction with hydrogen [8]. Recently, research on the deposition of Mo thin films using the atomic layer deposition (ALD) method has become increasingly popular as the integration of semiconductors has improved. Consequently, studies on precursors for depositing Mo thin films have been on the rise. In the processes prior to ALD, Mo-containing precursors were limited to a few selected and commonly known Mo precursors, such as $MoCl_5$, Mo $(CO)_6$, and alkylamine precursors. Recently reported precursor combinations for the deposition methods of $MoS_2$ thin films include $Mo(CO)_6$ and $H_2S$, $Mo(CO)_6$ and MeSSMe, and $MoCl_5$ and $H_2S$. However, these traditional Mo precursors encounter several problems [9,10]. It is difficult to deposit pure Mo thin films to replace tungsten either because they are very toxic materials or because Mo thin films containing impurities have been deposited. Mo alkylamine precursors can contain Mo with an oxidation state of +VI, which can cause problems during the deposition of Mo thin films. Mo alkylamine precursors, in which Mo has a more desirable oxidation state of +IV, are generally unstable and difficult to use. In addition, Mo alkylamine precursors are relatively temperature sensitive and can decompose at low temperatures. This can lead to the degradation of the Mo alkylamine precursor since relatively high temperatures are typically required to promote crystalline film growth [11–13]. To date, Mo thin films have failed to replace tungsten. Semiconductor manufacturers are currently facing a shortage of Mo precursors to replace tungsten. In this study, we investigate the properties of Mo thin films deposited using $MoO_2Cl_2$ precursors to determine the feasibility of developing a Mo thin film process applicable to metal wiring processes.

## 2. Experiments and Discussion

### 2.1. Experimental Method

2.1.1. Process Conditions for Molybdenum (Mo) Thin Film

In this study, a Mo thin film was deposited on a 12-inch wafer with silicon oxide. To measure the resistance of the Mo thin film, a bare wafer must be deposited on an insulator, where its resistance remains constant. The deposition equipment is an independently developed thermal ALD facility (model no: I2FIT-Mo) with Ar as purge and carrier gas and $MoO_2Cl_2$ and $H_2$ gas as precursors and reactants, respectively. The processing pressure was 11–20 torr, the deposition temperature was 600 °C to 650 °C, and the characteristics according to the temperature gradient were evaluated. The $MoO_2Cl_2$ precursor was solid. Unlike liquid precursors, the surface area in the canister changed with the increase in usage, thus a separate delivery system was organized to maintain the reproducibility of the process evaluation. Experimental conditions, such as process parameters, are shown in Table 2.

**Table 2.** Deposition experimental conditions (Mo).

| Parameter | Condition |
| :---: | :---: |
| Precursor | $MoO_2Cl_2$ |
| Reactant gas | $H_2$ |
| Purge gas | Ar |
| Carrier gas | Ar |
| Pressure | 11~20 Torr |
| Substrate | 12 inch Si(100)/$SiO_2$ (1000 Å) wafer |
| Process temperature | 600~650 °C |

The properties of the deposited Mo thin film were examined by the following analyses. The thickness of the deposited Mo thin film was calculated by measuring 49 points on a wafer using X-ray fluorescence (XRF). Rigaku Corporation's MFM310 model was used for XRF, and Mstech Corporation's 4-point probe was used to measure the resistivity of Mo

thin films. X-ray diffraction (XRD) (model: SmartLab) analysis confirmed the crystallinity of the thin film, and X-ray photoelectron spectroscopy (XPS) (model: K-Alpha+) analysis was performed to determine the degree of contamination due to O and Cl (impurities in the membrane). Surface analysis according to process variables was conducted to compare the surface roughness of Mo thin films by atomic force microscopy (AFM) (model: NX20) analysis. To confirm step coverage in the pattern, a Mo thin film was deposited on a wafer formed with a trench of an aspect ratio of 40:1 and step coverage was observed through transmission electron microscopy (TEM) (model: Tecnai $G^2$ F30 S-Twin) analysis.

### 2.1.2. Molybdenum-Nitride (MoN) Seed Layer Process Conditions

To deposit a pure Mo thin film, the oxide wafer surface was washed in deionized water for 10 min and then dried. It was expected that the deposition rate would be too low to grow a Mo thin film directly on the wafer, requiring a seed layer for initial nucleation. MoN thin films have been utilized as seed layers in previous studies as they have been demonstrated to be an antidiffusion barrier with excellent electrical and thermal stabilities, facilitating early nucleation [14,15]. Table 3 shows the deposition conditions of the MoN thin film deposited as a seed layer. It was deposited by thermal ALD using $NH_3$ as the reaction gas. ALD is an excellent method for depositing MoN thin films because the nature of the seed layer requires fine tuning.

**Table 3.** Deposition experimental condition (MoN).

| Parameter | Condition |
| --- | --- |
| Precursor | $MoO_2Cl_2$ |
| Reactant gas | $NH_3$ (10,000 sccm) |
| Purge gas | Ar (3000 sccm) |
| Carrier gas | Ar (100 sccm) |
| Pressure | 3~5 Torr |
| Substrate | 12 inch Si(100)/$SiO_2$(1000 Å) wafer |
| Process temperature | 600~650 °C |

### 2.2. Experimental Results and Discussion

### 2.2.1. Deposition of the Molybdenum Nitride (MoN) Seed Layer

Figure 1 is a graph showing the effect of the MoN seed layer on the growth of Mo thin film at a process temperature of 600 °C. The changes in thickness and resistivity with and without MoN deposition were verified. The deposition cycle of the Mo film was fixed at 280 cycles, and then the deposition cycle of the MoN film was varied. Mo thin films did not grow with the Mo deposition alone or ten cycles of MoN deposition. The growth of Mo with 20 cycles of MoN confirmed that a normal Mo thin film could only be grown by depositing a seed layer on the oxide wafer surface. To deposit Mo ALD thin films, it is critical to understand the mechanism by which the crystal nuclei on the wafer surface are first created, followed by the crystals' growth [16]. The rates of crystal nucleation and growth are kinetic variables that determine the formation of the final crystalline material and the crystal size distribution. Among them, the stage of crystal nucleation is the main limiting factor for the kinetic reaction rate of crystallization. Crystal nucleation is divided into primary nucleation and secondary nucleation. Primary nucleation is the aggregation of particles (i.e., molecules, atoms, or ions) bound to a precursor to produce a crystal nucleus and is divided into homogeneous nucleation and heterogeneous nucleation. Homogeneous nucleation is a phenomenon where spontaneous nucleation occurs on the wafer surface and by precursor molecules. Conversely, heterogeneous nucleation is a phenomenon where crystallization is induced by impurities or foreign surfaces. The impurities or foreign surfaces are often referred to as seed materials; the general catalytic effect of these seed materials is to reduce the energy required for nucleation, thereby promoting an increase in

the rate of nucleation. Secondary nucleation is a self-catalyzed reaction, in which a crystal nucleus is produced in the presence of a homogeneous seed crystal material [17–20]. It is expected that the MoN thin film is acting as a self-catalyst in the generation of Mo thin films. Figure 2 shows the cross-sectional TEM analysis of a MoN/Mo thin film. An overall increase in thickness is observed with increasing MoN deposition cycles. We observe a thickness trend that is almost similar to the XRF thickness gauge results, as shown in Figure 1. The chemical Equations (1) and (2) between $MoO_2Cl_2$ precursor and $NH_3/H_2$ reaction gas are as follows:

$$MoO_2Cl_2 \text{ (g)} + NH_3 \text{ (g)} \rightarrow MoN \text{ (s)} + 2HO \text{ (g)} + HCl_2 \text{ (g)} \tag{1}$$

$$2MoO_2Cl_2 \text{ (g)} + H_2 \text{ (g)} \rightarrow 2Mo \text{ (s)} + 2O_2 \text{ (g)} + 2HCl_2 \text{ (g)} \tag{2}$$

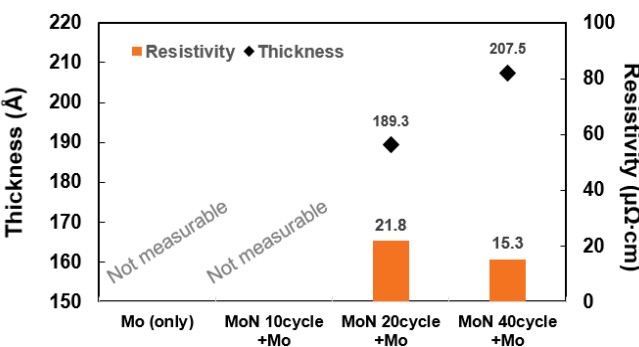

**Figure 1.** Identification of the MoN seed layer's influence on the Mo thin film (@600 °C).

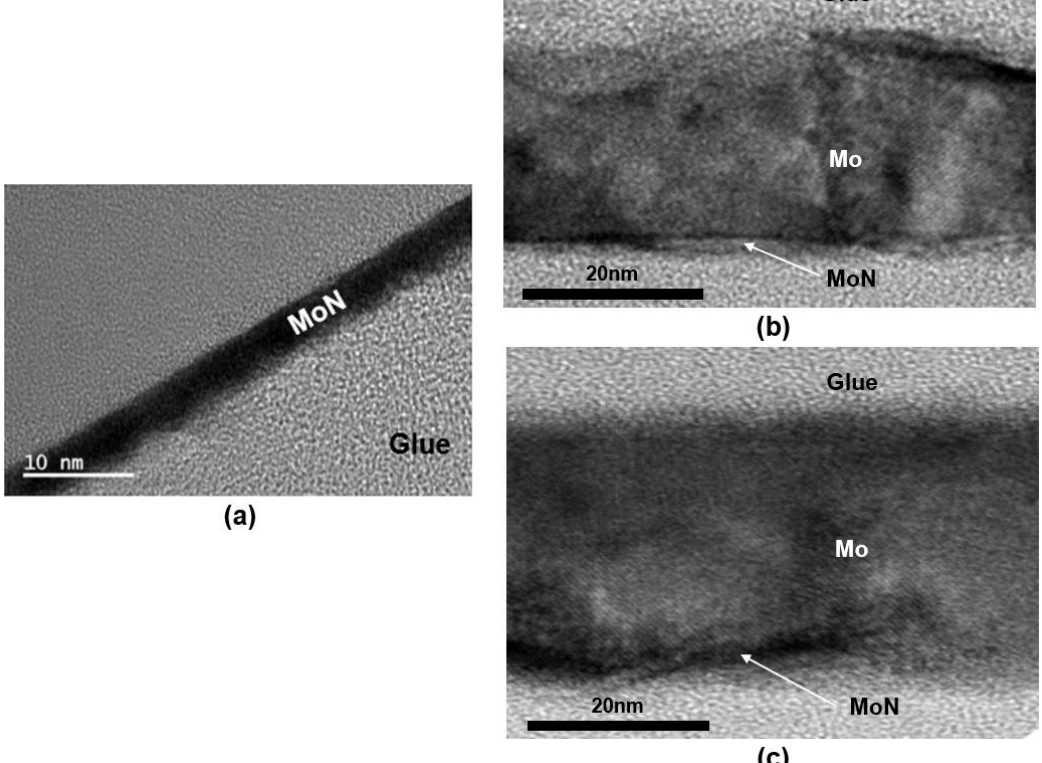

**Figure 2.** TEM cross-sectional profiles of (**a**) MoN (40 cycles) (**b**) MoN (20 cycles) + Mo (280 cycles), (**c**) MoN (40 cycles) + Mo (280 cycles).

### 2.2.2. Evaluating the Fundamental Properties of Molybdenum (Mo) Thin Films

Figure 3 shows the XRD analysis results of the Mo thin film as a function of the MoN deposition cycle. In the figure, MoN ((110) and (200)) peaks were observed at 2θ values of 36° and 45.5°, indicating that MoN thin films were deposited successfully. Furthermore, the largest (110) peak at a 2θ value of 41° was observed, suggesting that the deposited Mo thin films form a cubic crystal structure [21]. Typically, Mo thin films have peaks of (200) and (211) along with (110), corresponding to its secondary phase.

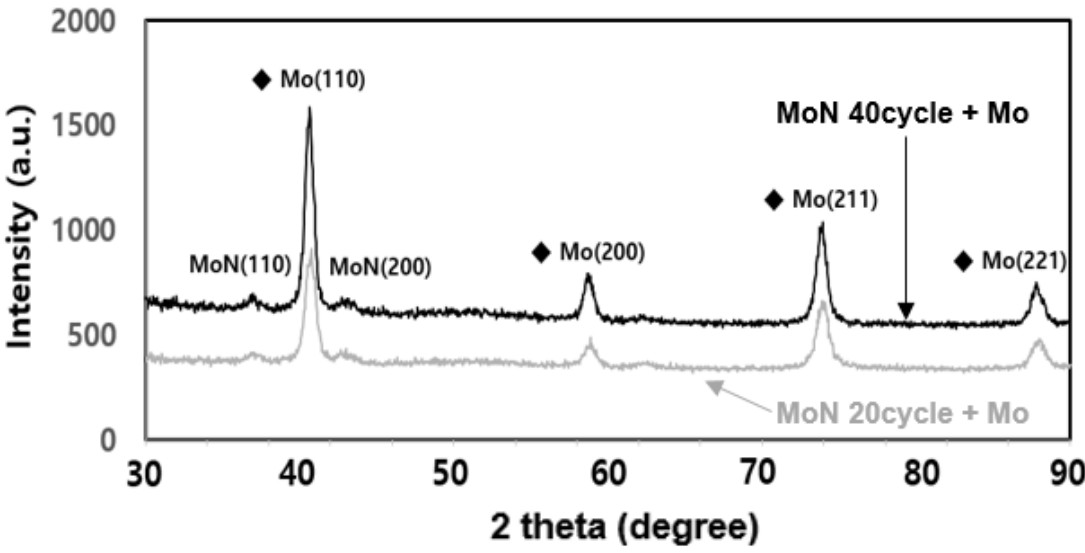

**Figure 3.** X-ray diffraction analysis results of MoN/Mo (280 cycles) thin films.

Factors affecting the reactivity of the precursor and reactive gas in thermal ALD processes include process temperature, the reaction time of the precursor and wafer, and the amount of gas reacting with the precursor, which are the process parameters with considerable impact on deposition characteristics [22,23]. Figure 4 shows the variation of thickness and resistivity of MoN/Mo thin films with each process parameter. Figure 4a shows the trend of thickness and resistivity when the source feeding time is increased from 1 to 5 s. It shows that as the source feeding time increases, the thickness of the film increases and the resistivity decreases. Figure 4b displays the trend of thickness and resistivity when the reactant $H_2$ flow rate is increased from 5000 to 15,000 sccm. It shows that as the reactant $H_2$ flow rate increases, the thickness of the film increases and the resistivity decreases. Moreover, Figure 4c shows the trend of thickness and resistivity when the source carrier flow rate is increased to 100 sccm. It describes that as the source carrier Ar flow rate increases, the thickness of the film increases and the resistivity decreases. It was found that increasing the amount and time of reactants reacting with the wafer, increased the thickness of the Mo film but did not improve its resistivity properties. The deposition rate increases as the amount of source and reactant gas supplied within the chamber increases with all deposition conditions being identical. If the residual source and reactant gases on the wafer surface do not increase and thus are not sufficiently purged, the ideal self-limiting reaction may not occur, and two or three layers of atoms may be deposited instead of one. The resistivity is the intrinsic resistance of a substance. Although the value of the resistivity is known for pure substances, resistivity can be calculated using various methods for mixtures. As in this experiment, if a surface resistance meter with a four-point probe is used, the resistance can be computed by multiplying resistance with thickness. The formula for computing the resistivity is shown in Equation (3) below.

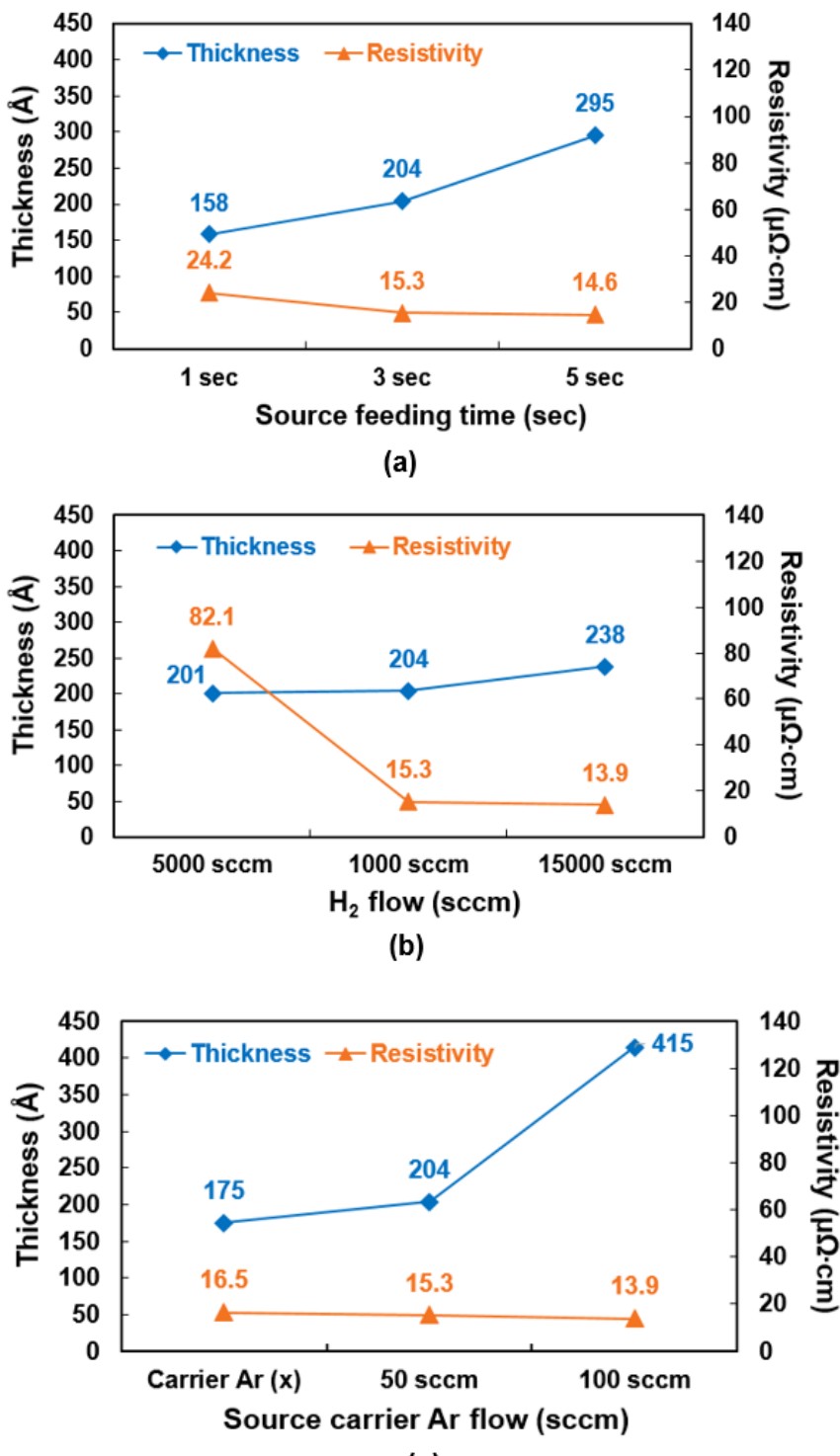

**Figure 4.** Thickness and resistivity variation by process parameter for MoN (40 cycles)/Mo(280 cycles) thin films. (**a**) Change in thickness and resistivity due to change in source feeding time, (**b**) Change in thickness and resistivity due to change in reactant $H_2$ flow, (**c**) Change in thickness and resistivity due to change in source carrier Ar flow.

$$\text{Resistivity } (\Omega cm) = \text{Resistance } (\Omega/sq) \times \text{Thickness (cm)} \qquad (3)$$

∗  Resistance = $\Omega$ × Correction Factor = $\Omega$/sq
∗  Correction Factor = 4.532 × 1 × 1

- Sample size coefficient for the sample with a diameter greater than or equal to 40 mm = 4.532
- Film thickness coefficient at approximately 400 μm or less thickness = 1
- Temperature coefficient at 23 °C = 1

Mo thin films benefit from their lower resistivity, which increases their use as gate electrodes in MOSFET [24–27].

XPS measurements were performed to analyze the composition of Mo in the thin film according to the process parameters (Figure 5). In all three conditions, about 5% to 6% of the O component was found in addition to the Mo component. However, unlike the case demonstrated in Figure 4, the trend of increasing concentration of the impurity O in the film and increasing resistivity is not confirmed.

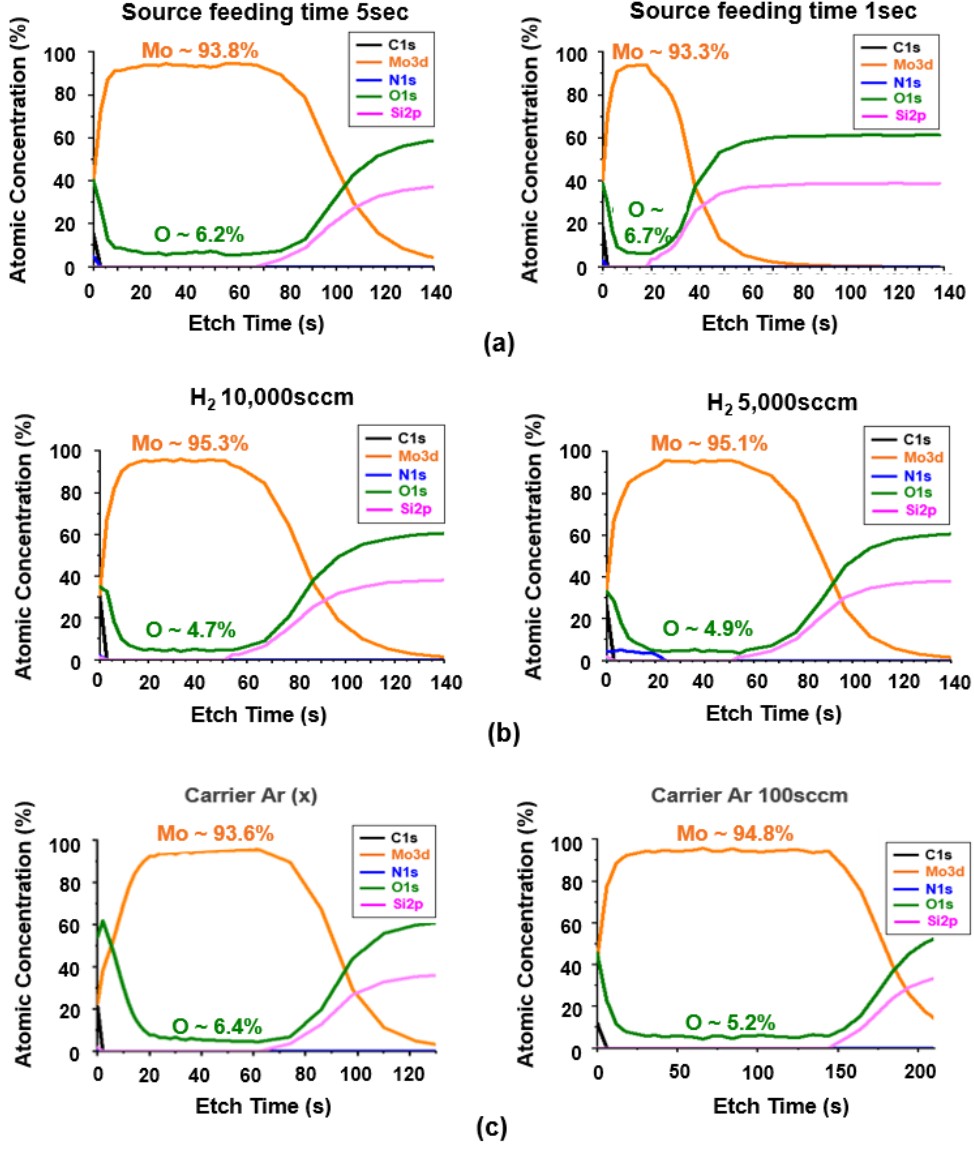

**Figure 5.** XPS analysis results of MoN (40 cycle)/Mo thin film by process parameters: (**a**) Mo/O concentration change in the film due to change in source feeding time, (**b**) Mo/O concentration change in the film due to change in reactant H$_2$ flow, and (**c**) Mo/O concentration change in the film due to source carrier Ar flow.

According to the literature, one of the most important process variables that can improve the resistivity of Mo thin films is the deposition temperature [28]. A process

temperature that is sufficient for the substitution reaction between $TiCl_4$ precursor and $NH_3$ reaction gas is required to activate the self-limiting reaction. Generally, the deposition temperature of Mo thin films deposited on the $MoCl_5$ precursor using the CVD method is between 500 °C and 800 °C [29]. In this study, the deposition characteristics in the temperature range of 600 °C to 650 °C were investigated in detail, considering the ALD deposition characteristics, process, and equipment margins. Carrier Ar, which is an important parameter for deposition characteristics, was fixed at 150 sccm, and the reaction gas $H_2$, was fixed at 15,000 sccm.

Figure 6 shows the variation of deposition rate and resistivity with increasing deposition temperature. As the deposition temperature increased from 600 °C to 650 °C, the deposition rate per cycle increased from 0.731 Å to 0.787 Å, and the resistivity characteristics of the Mo film improved to 12.9 μΩ·cm. The lower resistivity value with the increasing deposition temperature implied that the electrical properties of the Mo thin film were improving with the increasing deposition temperature. It is known that resistivity is affected by impurities in the film, film thickness, microstructure, composition ratio, etc., and the value varies depending on the formation conditions of the film. A previous report has shown that for most metal thin films, the average grain size increases and the resistivity improves with increasing deposition temperature [30]. Particularly, studies have shown that an increase in deposition temperature of 100 °C improves the resistivity characteristics by ~40% [31] for TiN thin films used as barrier metals.

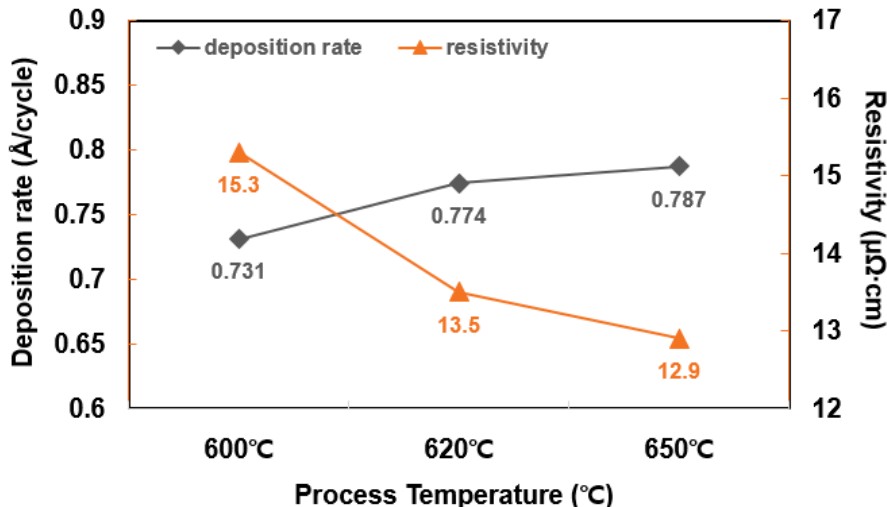

**Figure 6.** Variation of deposition rate and resistivity of MoN (40 cycles)/Mo (280 cycles) thin films with increasing process temperature.

In the case of Mo thin films, which are crystalline conductive films, the greater the crystallinity, the denser the structure. The crystallinity of the Mo thin film was increased by increasing the process temperature to 650 °C. In this study, an aluminum nitride heater was used, and the maximum temperature was set to 650 °C because it is difficult to control the heater beyond that temperature owing to the heater characteristics. Therefore, it has low thermal damage and considerable adhesion even at high temperatures during the process, which are considered favorable characteristics for their use as a gate conductive film [32]. Figure 7 is the analysis result of XRD measured at deposition temperatures of 600 °C to 650 °C to confirm the crystal structure of the MoN/Mo thin film. The largest (110) diffraction peak at 2θ = 41° was also observed at 620 °C and 650 °C, indicating that it formed a cubic structure. As the process temperature increased from 600 °C to 650 °C, the intensity of Mo (110) and (200) diffraction peaks tended to decrease, and the intensity of Mo (211) and (221) diffraction peaks tended to increase. The reason for this was that the deposition temperature increased and the thermal driving force that could form the Mo thin film increased, changing the structure of the thin film [33,34].

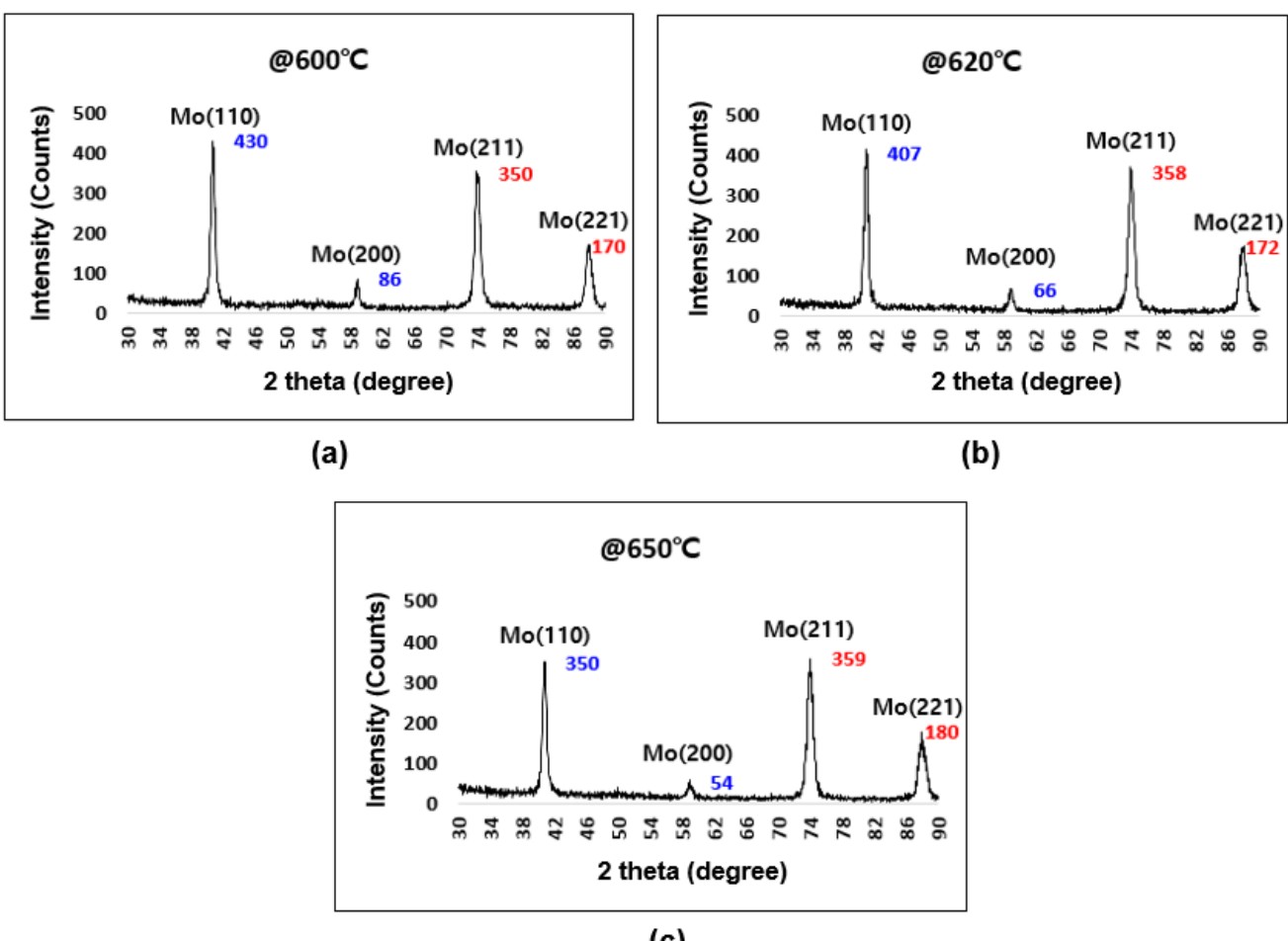

**Figure 7.** X-ray diffraction analysis of MoN (40 cycles)/Mo (280 cycles) thin films with increasing process temperature. (**a**) process temperature 600 °C, (**b**) process temperature 620 °C, (**c**) process temperature 650 °C.

It is known that the morphology of the surface is usually improved when the grain size of the particles constituting the thin film is small. Therefore, increasing the nuclear density can improve the morphology of the surface [35,36]. The higher the nuclear density, the greater the amount that can change into a solid state, resulting in the occurrence of nuclei at multiple sites. In other words, nuclei with different orientations are produced in several places, forming several small grains. Nuclear density is largely affected by substrate type, pressure, and temperature and is usually inversely proportional to wafer temperature [34]. In this study, AFM analysis was performed to confirm the surface morphology of Mo thin films deposited at 600 °C or above to confirm the effect of deposition temperature on surface morphology (Figure 8). Surface roughness (Rq) at 600 °C was 0.499 nm; Rq at 620 °C was 0.513 nm; and Rq at 650 °C was 0.560 nm. These results confirmed that the surface roughness increased further with increasing temperature. As the deposition temperature increased, the columnar shape became more developed, and the grain size increased. Thus, an increasing Rq might also be expected.

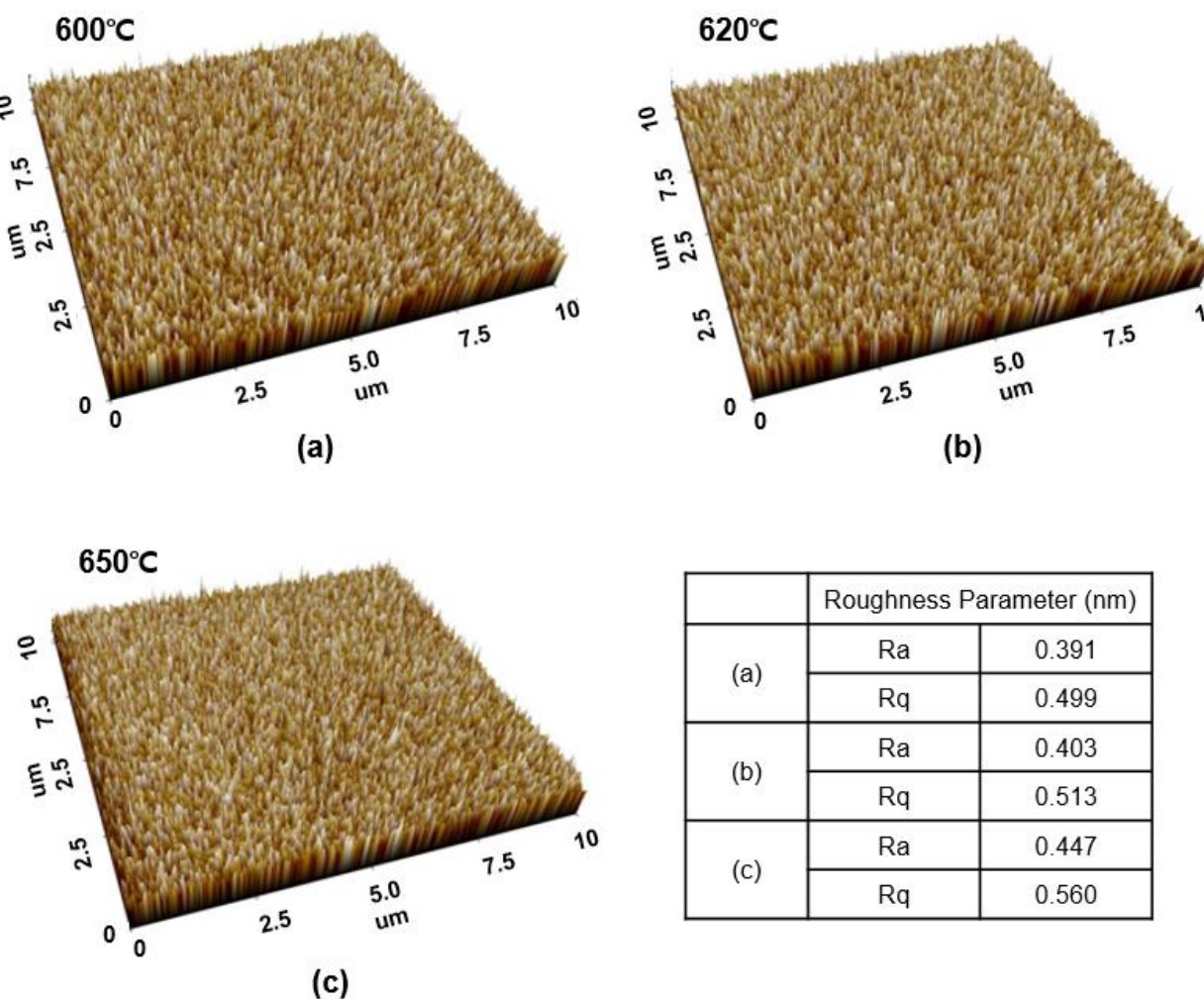

**Figure 8.** Surface morphology analysis results of MoN (40 cycles)/Mo (280 cycles) thin film with increasing process temperature. (**a**) process temperature 600 °C, (**b**) process temperature 620 °C, (**c**) process temperature 650 °C.

### 2.2.3. Molybdenum (Mo) Deposition on Patterned Wafers

In our experiments, the maximum process temperature of the ALD facility that can be used was 650 °C, and the previous experiments confirmed a deposition rate of 0.787 Å per cycle and a resistivity of 12.9. With the miniaturization of semiconductor patterns, improving step coverage characteristics is becoming increasingly important; particularly, improving step coverage characteristics in high aspect ratio patterns is a crucial challenge. To measure step coverage under the condition of a process temperature at 650 °C, a Mo thin film was deposited on a line trench pattern with an aspect ratio of 40:1 (Figure 9). The depth of the pattern used for step coverage measurement was 2400 nm, and the CD was 60 nm. We measured the top, middle, and bottom sections and confirmed 97% step coverage. Generally, as the deposition temperature increases, the thermal decomposition increases, resulting in a decrease in step coverage. Thus, increasing the step coverage is feasible if the amount of precursor is reduced and the process conditions are adjusted to enhance the purge.

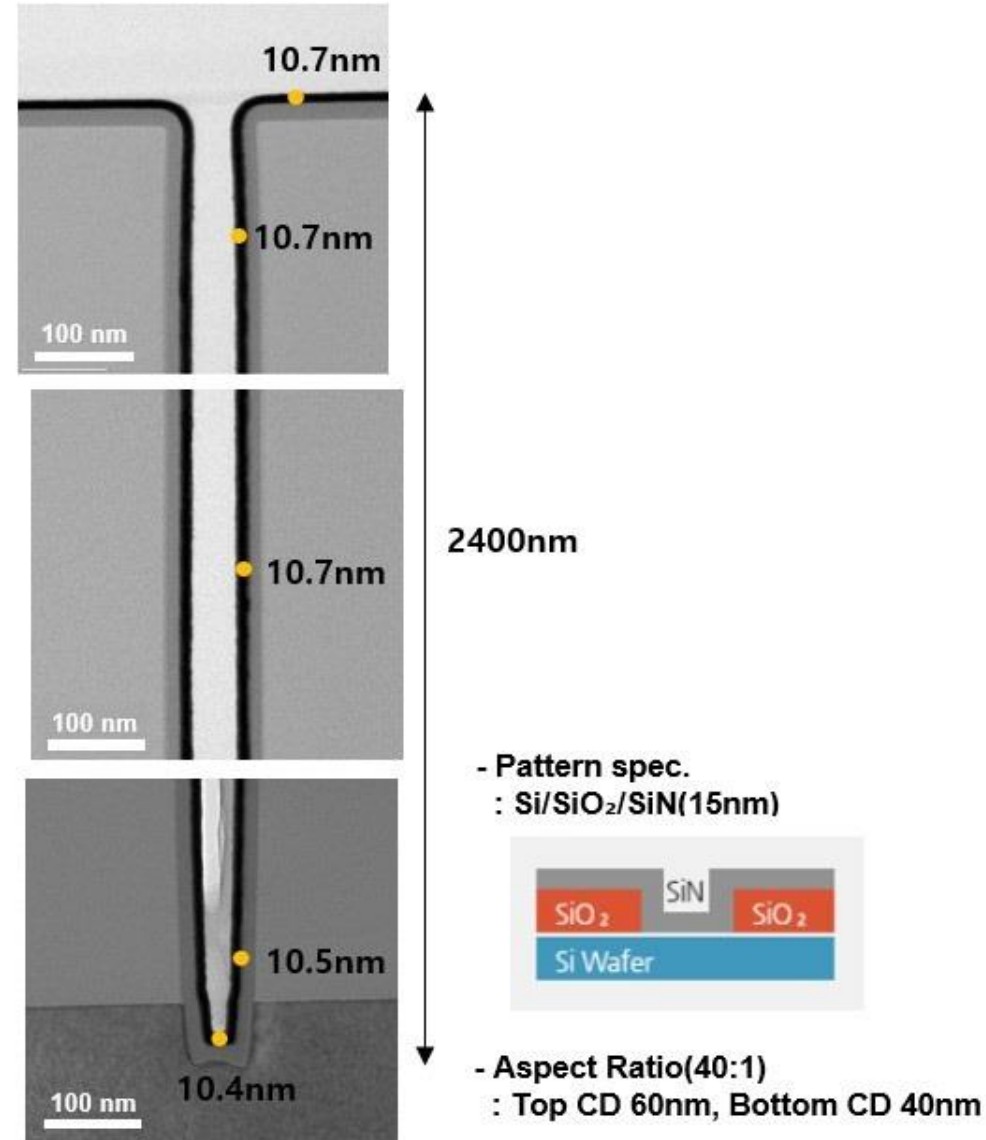

**Figure 9.** Step coverage results for 40:1 pattern of MoN (40 cycles)/Mo (140 cycles) thin films at a process temperature of 650 °C.

## 3. Conclusions

In this paper, we conducted a study of manufacturing Mo thin films using a solid precursor, $MoO_2Cl_2$. In general, solid precursors have inconsistent vapor pressures compared to that of liquid precursors, making it difficult to obtain process characteristics. Herein, we developed a delivery system that kept the vapor pressure of the solid precursor constant and supplied it to the chamber. Consequently, we successfully deposited Mo thin films at a process temperature of 600 °C or above with our independently developed thermal ALD system using $MoO_2Cl_2$ precursors. While depositing Mo thin films, a certain thickness of the MoN layer was required for initial nucleation on the wafer. In the fabrication of Mo thin films deposited by thermal ALD, we investigated the differences in fundamental physical properties because of the difference in deposition temperature and found that the deposition rate increased, and the resistivity decreased as the deposition temperature increased. The deposition rate per cycle of the Mo thin film deposited at 650 °C was found to be 0.787 Å, and the step coverage was identified as 97% (with an aspect ratio of 40:1). Further, this study found that an increase in the deposition temperature during Mo

deposition results in greater thermal driving force, which modifies the structure of the Mo film and increases crystallinity and thereby contributes to the resistivity drop.

**Author Contributions:** B.-J.L.; Conceptualization, Data curation, Writing—review & editing, K.-B.L.; Resources, M.-H.C.; Methodology, D.-W.S.; Formal analysis, Investigation, J.-W.C.; Methodology, Software. All authors have read and agreed to the published version of the manuscript.

**Funding:** This research received no external funding.

**Institutional Review Board Statement:** Not applicable.

**Informed Consent Statement:** Not applicable.

**Data Availability Statement:** Not applicable.

**Acknowledgments:** This study was conducted with Hanwha Corporation's own funding.

**Conflicts of Interest:** The authors declare no conflict of interest.

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
