# Peer review of "Study on the Deposition Characteristics of Molybdenum Thin Films Deposited by the Thermal Atomic Layer Deposition Method Using MoO2Cl2 as a Precursor"

_coatings, doi:10.3390/coatings13061070_

Round 1

Reviewer 1 Report

This article systematically investigates the various parameters of ALD-grown conductive Mo films. The work presented in the paper is detailed and comprehensive, and the research direction aligns well with the Coating journal. However, compared to the quality of the research itself, the image processing and table formatting in the paper are rough and messy. It is suggested to make modifications to ensure that the paper can be accepted by the Coating journal.

The quality of English language is well.

Reviewer 2 Report

The authors present the results on manufacturing molybdenum (Mo) thin films by thermal atomic layer deposition method using MoO2Cl2 precursor (solid) to determine the feasibility of developing a Mo thin film process applicable to metal wiring processes. In my opinion, the manuscript is an interesting and relevant topic. However, the manuscript is non-organized, the abstract, the experimental results have several drawbacks. Finally, in my opinion, I suggest that this article should not be accepted as presented. Therefore, I suggest that the following major revisions be considered for acceptance.

1.- Abstract must contain the major aspects of the entire paper in a prescribed sequence that includes: 1) the overall purpose of the study and the research problem you investigated; 2) the basic design of the study; 3) trends found as a result of your analysis; and, 4) a brief summary of your interpretations and conclusions

2.- The introduction is not clear on the relevant aspect of the investigation, results, and future perspectives.

3.- Abstract and introduction: What does the nomenclature MoO2Cl2, H2, W, H2S, MeSSMe, MoCl5, WF6, Mo (CO)6, MoS2? please define acronyms.

4.- To improve the organization in the results, I consider adding a new section where all the materials and methods used in the research are described, this new section can be calling “Materials and Methods”.

5. In Section Experimental Results and Discussion. Please, just focus on results and discussion, not mentioning all obtention of the material (just essential information of the experimentation).

6. The experimental design is unknown. How many executions were performed to obtain those values shown in Figure 8 (Ra).

7. It is not clear that the conclusions are supported by the results, besides the abstract mentioned that depositing high-quality Mo thin films, where does it reflect in the results?

8.The references should be updated.

Moderated editing of English language 

Reviewer 3 Report

The manuscript is well written. However, before it can be published the following problems should be addressed:

The introduction is very specific, in the begining one paragraph must be added, and discus about the different coating techniques. In this case use new articles, for example:

https://doi.org/10.1016/j.surfcoat.2023.129500

https://doi.org/10.1016/j.surfcoat.2022.129133

https://doi.org/10.1007/s12540-020-00692-y

https://doi.org/10.1016/j.diamond.2019.04.007

Please try to discus about different type of coatings such as PACVD, Electrodeposion, electroless ass I mentioned, and bold out the benifit of atomic layer deposition comparing to them.

The english needs minor revision.

Reviewer 4 Report

In this work, the authors deposited the Mo thin films using MoO2Cl2 precursors. The effects of deposition characteristics (e.g. seed layer thickness, process parameters, and temperature) on the properties of Mo thin films were investigated. Detailed experiments were conducted and the corresponding mechanism was elucidated. In my point of view, this work can be considered for publication. However, there are still things to be improved. For instance, the following issues need to be addressed:

1) In introduction, the authors write: “As a next-generation semiconductor material, molybdenum (Mo) thin films possess excellent physical properties of extremely high melting point, low thermal expansion coefficient, low resistivity, and high thermal conductivity. Consequently, it is increasingly used to manufacture semiconductor devices for diffusion barriers, electrodes, photomasks, power electronic device substrates, and low-resistivity gates and connectors...Because of their low resistivity, transition metals, such as Mo, are widely used as metal electrodes in solar cells or semiconductors, where energy loss must be minimal [1-3].” The general reference list seems a bit thin, considering the evolution in the field within the recent years. To give the readers a much broader view, the potential application of Mo for manufacturing nitride semiconductor LEDs, such as Laser & Photonics Reviews 2023, 17, 2200455 (https://doi.org/10.1002/lpor.202200455); Optics Express 27(12), A669 (2019); Optics Letters 47(5), 1291-1294 (2022), etc. should be added, so that the readers can be clear about the state-of-the-art of this topic.

2) Why is the resistivity of MoN 20 cycles + Mo larger than that of MoN 40 cycles + Mo?

3) The mechanism that the resistivity of Mo thin films was decreased when increasing the source feeding time and gas flow should be further discussed.

4) Did the authors have tried to characterize the concentration of the impurity O in Mo thin films when increasing the deposition temperature?

5) In Fig. 6, we can note that the changing amplitude of disposition rate and resistivity is reduced with the increase of temperature. What is the reason for that?

Reviewer 5 Report

The authors studied the manufacturing Mo thin films using a solid precursor MoO2Cl2 deposited by thermal atomic layer deposition method. They optimised the growth conditions by varying the growth time, the Ar and H2 flow rates, and then the growth temperature to reduce the resistivity of Mo layers. This work has some merit; however it should be improved by following the reviewer comments:

1/ What are the conditions on the Ar and NH3 flow gas in table 2? What kind of oxide used as a wafer?

2/ From XRD, the authors can determine the dislocation density of the Mo layer to correlate it with its resistivity.

3/ The authors should add more explanation for the reason of the increase of Mo layer thickness with the increase of the Ar or H2 flow gas and its relationship with the kinetic growth.

4/ Figure 5c should be corrected (add the flow rate of Ar).

5/ The authors should precise the used time and flow rate of Ar and H2 when they assessed the temperature effect on the resistivity of the Mo layer.

6/ A comparison with W layer properties should be done especially this is commonly used for electrodes and the authors expected to replace the existing metal gate and tungsten thin films.

 Minor editing of English language is required.

Round 2

Reviewer 2 Report

The authors have improved the quality of their writing. However, in my opinion, the manuscript can be accepted after minor revisions in text editing.

Minor editing 

Reviewer 3 Report

The introduction is very weak and the authors make no attempt to correct it based on reviewer advice. The should be a logical and scientific comparision between the coatings it was mentioned in previous comment. The author makes no attempt to improve it as it was suggested. This will cause a question for readers and manuscript cannot be pulished before fixing this problem

Some of the refrences are not relevant. Almost more than 10 refrences changed after revision. How is this possible? Double check them carefully

The number of analysis are not enough to do discussion

The elemen distribusion should be added to TEM otherwise you cannot mark Mo on image.

The schematic discussing the mechanism must be added to the manuscript.

It is suggested to add XPS for discusing the nature of chemical bonding.

There is no error bar in any results. Add it and explain how you calculate error bar.

English needs minor revision

Reviewer 5 Report

The authors corrected the manuscript following the reviewer comments. This work can be considered for publication in Coatings.

Minor editing of English language is required.

Author Response

Finally, the introductory part has been corrected. The advantages of the ald deposition method were introduced. Thank you for your review.